# Electrospinning Inorganic Nanomaterials to Fabricate Bionanocomposites for Soft and Hard Tissue Repair

**DOI:** 10.3390/nano13010204

**Published:** 2023-01-02

**Authors:** Jie Cui, Xiao Yu, Yihong Shen, Binbin Sun, Wanxin Guo, Mingyue Liu, Yujie Chen, Li Wang, Xingping Zhou, Muhammad Shafiq, Xiumei Mo

**Affiliations:** 1State Key Laboratory for Modification of Chemical Fibers and Polymer Materials, Shanghai Engineering Research Center of Nano-Biomaterials and Regenerative Medicine, College of Biological Science and Medical Engineering, Donghua University, Shanghai 201620, China; 2College of Science, Donghua University, Shanghai 201620, China; 3Department of Chemical Engineering, Faculty of Engineering, Graduate School, Kyushu University, 744 Motooka, Nishi-Ku, Fukuoka 819-0395, Japan; 4Department of Biotechnology, Faculty of Science and Technology (FOST), University of Central Punjab (UCP), Lahore 54000, Pakistan

**Keywords:** tissue engineering, inorganic materials, electrospinning, nanofibers, scaffolds

## Abstract

Tissue engineering (TE) has attracted the widespread attention of the research community as a method of producing patient-specific tissue constructs for the repair and replacement of injured tissues. To date, different types of scaffold materials have been developed for various tissues and organs. The choice of scaffold material should take into consideration whether the mechanical properties, biodegradability, biocompatibility, and bioresorbability meet the physiological properties of the tissues. Owing to their broad range of physico-chemical properties, inorganic materials can induce a series of biological responses as scaffold fillers, which render them a good alternative to scaffold materials for tissue engineering (TE). While it is of worth to further explore mechanistic insight into the use of inorganic nanomaterials for tissue repair, in this review, we mainly focused on the utilization forms and strategies for fabricating electrospun membranes containing inorganic components based on electrospinning technology. A particular emphasis has been placed on the biological advantages of incorporating inorganic materials along with organic materials as scaffold constituents for tissue repair. As well as widely exploited natural and synthetic polymers, inorganic nanomaterials offer an enticing platform to further modulate the properties of composite scaffolds, which may help further broaden the application prospect of scaffolds for TE.

## 1. Introduction

Tissue engineering (TE) is an emerging field of bioengineering, which has witnessed continuous development recently. The TE combines techniques and principles from multidisciplinary research disciplines, such as engineering, cell biology, and materials science to create native-like artificial tissues [1]. Moreover, TE is revolutionizing healthcare by providing on-demand, artificially developed tissues and organs for regenerative medicine. Intelligent multifunctional scaffolds provide instructional cues for the precise manipulation of cells in vitro and in vivo as well as to drive their assembly into artificial tissues to either replicate the in vivo microenvironment for disease and development-related studies or to assemble platforms for in vivo implantation [2]. A myriad of scaffolds based on an array of materials has been explored for TE applications. In addition, a series of biomaterials, such as natural and synthetic polymers [3,4,5], ceramics [6,7,8], silk proteins [9,10], alginates [11], chitosan [12,13], cellulose [14,15], and bioactive molecules [16,17] has been explored for different TE disciplines. These scaffold materials have shown promise for the regeneration of different types of tissues, including bone, cartilage, skin, and tendon [18,19,20] through cellular and signaling stimulation.

In addition to natural and synthetic polymers, inorganic materials, such as metal oxides (MOs), metal nanoparticles (NPs), and carbon-based nanomaterials (NMs) are being intensively studied for TE applications [21]. Layered silicate nanoclays have been widely pursued for dermatological and musculoskeletal applications [22,23]. Similarly, carbon-based NMs have been exploited as fillers for TE applications owing to their chemical stability, low coefficient of friction, good mechanical properties, heat and wear resistance, high electrical conductivity, and hardness [24,25,26]. On the other hand, MOs, including bioceramics, bio-glasses (BGs), and magnetic NPs have also been exploited for TE. Of these, bioceramics have been shown to induce biomineralization due to their excellent osteo-conductivity, chemical resistance, and durability. Bioceramics can be further classified as biologically inert, bioactive, or bioresorbable, which are mainly based on their interaction with the host tissues in vivo [27]. While biologically inert ceramics are physically and chemically stable and do not interact with the tissues, bioactive ceramics can repair, replace and regenerate tissues. On the other hand, bioresorbable ceramics gradually degrade in vivo without inducing obvious toxicity risks. Metal NPs are also widely exploited in TE due to their high stability and ease of synthesis. Different types of metal NPs, such as gold (Au), silver (Ag), iron (Fe), aluminum (Al), nickel (Ni), copper (Cu), strontium (Sr), and zirconium (Zr) [28] have been shown to play a pivotal role in regulating cellular behavior as well as promoting tissue regeneration. Since inorganic NMs exhibit unique physico-chemical and mechanical properties, their introduction into TE scaffolds may impart bio-functionality as well as improve elasticity and resistance to mechanical stress. Consequently, bio-scaffolds comprised of inorganic/organic hybrids may help realize customized biomechanical properties as well as sufficient bioresorbability [29].

Although polymeric scaffolds may induce tissue repair, the addition of inorganic NMs may further impart complementary functionalities to the scaffold; thereby, further broadening their application prospects. For instance, the addition of inorganic NPs may improve the osteogenic properties for bone tissue regeneration, antimicrobial activity and angiogenesis for wound healing, and electrical conductivity and anti-oxidative properties for nerve repair [30,31].

While inorganic NMs have been widely explored for TE, there is an acute lack of comprehensive reports detailing their use in electrospun nanofibers and organic–inorganic hybrids for different types of regenerative medicine and TE disciplines. We, herein, summarize various application forms and routes of inorganic NMs in electrospinning technology. Specifically, there are two major categories, including preparation of single inorganic nanofibers by electrospinning technique and the application of inorganic NMs to afford organic–inorganic composite nanofibers for TE. It is worthy to mention here that previously excellent reviews have been published about therapeutics and antibiotics loaded organic/inorganic nanofibers as well as inorganic NPs loaded composite nanofibers, however, which mainly focused on drug delivery and TE for nerve and skin regeneration [32,33].As compared to the previous papers, this manuscript provides a more comprehensive overview of the use of inorganic NMs in TE by electrostatic spinning, including both the repair of soft and hard tissues. Additionally, we surveyed recent articles about inorganic components doped electrospun nanofibers, which were also mentioned in previous papers (Figure 1). In addition, we also focus on the fabrication of pure inorganic nanofibres and their applications as well as the application of pure inorganic nanofibres as matrices to prepare composite scaffolds, which is scarce in the literature. For the application of inorganic NMs we have a clearer classification in two broad categories: (i) electrostatically spun pure inorganic nanofibres and (ii) electrostatically spun organic/inorganic composite nanofibres, which authors recognize as a novel aspect of this manuscript (Figure 1). 

## 2. Electrospun Inorganic Nanofibers for Tissue Engineering Applications

Electrospinning technology uses a high voltage power supply to charge the surface of a polymer solution or solute under the influence of an applied electric field, which then accelerates the flow of the jet towards a collector of the opposite polarity. As the electrostatic gravitational force between the oppositely charged liquid and the collector, while electrostatic repulsion among the similar charges in the liquid becomes stronger, the solution forms a Taylor cone at the tip of the propeller. Once the strength of the electric field exceeds the surface tension of the liquid, the fiber jet is eventually ejected from the Taylor cone, while the solvent evaporates during the jet stroke and the solid polymer fibers are deposited on the collector to afford micro/nanofibers [34] (Figure 1A). The nature of the polymer itself, the magnitude of the applied electric field, spinneret-to-collector distance, temperature, and other environmental factors may influence the morphology and diameter of the nanofibers [35]. Depending upon the collector as well as its rotational speed, randomly-oriented or aligned micro/nanofibers can be fabricated to control the overall mechanical properties as well as the biological response of the scaffold.

Another electrostatic spinning technology of note is needle-free electrostatic spinning, which is characterized by the fact that it does not employ needle nozzles, but rather uses a free liquid surface where Taylor cones are randomly generated to produce several polymer jets [36]. Consequently, a variety of needleless spinnerets have been designed for the fabrication of scaffold materials, such as spiral coils, yarns, cylinders, discs, balls and wires [37,38,39,40,41,42,43,44,45,46] (Figure 1B).

In recent decades, electrospinning technology has attracted the considerable interest of the research community due to its simplicity, versatility, and cost-effectiveness [47,48,49,50]. Different types of electrospun nanofibrous scaffolds are being exploited for various TE disciplines. In addition, an array of materials has been employed for the mass production of continuous nanofibers with appropriate mechanical properties [50,51]. Being one of the most common biofabrication techniques, electrospinning has been widely used to produce micro- and nano-fibers manifesting one-dimensional (1D) to three-dimensional (3D) morphologies [52]. These micro/nanofibers can mimic the characteristics of the natural extracellular matrix (ECM), which may also have implications for regenerative medicine and TE applications.

In addition to its ability to afford micro/nano-fiber scaffolds with tailorable porosity and pore size [53,54,55], the simplicity and cost-effectiveness of electrospinning technique renders it as a potential polymer processing technique for TE applications [56]. Appropriate porosity and pore size of electrospun fibers may leverage essential cues to influence multiple cellular effects, including adhesion, proliferation, differentiation, and migration. Moreover, electrospun nanofibers may afford temporal and spatial release of biological cues, such as growth factors (GFs), peptides, and nucleic acid therapeutics to further facilitate tissue repair [57,58,59,60,61,62].

The physicochemical aspects of scaffolds, such as mechanical properties, biocompatibility, and degradability may be tailored by using appropriate materials during electrospinning, which may further broaden the applicability of electrospun scaffolds. Until now, a series of materials has been electrospun, which include natural and synthetic polymers as well as inorganic NMs. The latter may further be customized to afford the sustained and controlled release of therapeutic ions for tissue repair. While comprehensive reviews have already been devoted to polymer-based electrospun scaffolds [51,63], there is an acute scarcity of the reports about inorganic NMs-based nanofibrous scaffolds. Recently, the authors of this study and others have designed nanofibrous scaffolds solely based on inorganic NMs, which have shown potential to induce functional tissue repair as well as induce biomineralization and neovascularization in vitro and in vivo [64]. Therefore, we surmised to review inorganic NMs-based scaffolds for TE applications.

### 2.1. Electrospinning of Pure Inorganic Nanofibres

Herein, we will be discussing the research reports related to the fabrication of nanofibrous scaffolds either based on pure inorganic NMs or the combination of inorganic NMs along with metal ions, natural polymers, or synthetic polymers. We have also enumerated the applications of these nanofibrous scaffolds for TE.

#### 2.1.1. Bioactive Glass-Based Electrospun Scaffolds

Among the inorganic components, BGs are a group of inorganic NMs that have been widely exploited for the treatment of bone defects, primarily owing to their ability to promote bone repair through therapeutic ions release or the formation of a superficial layer of hydroxyapatite (HAp) upon exposure to the physiological fluids [65,66]. This surface layer resembles the chemical composition and structure of bone minerals and therefore plays a key role for osteo-inductivity and interaction with the surrounding bone tissue. The BGs belong to a well-known class of synthetic bone replacement materials, which have been harnessed to mimic the 3D nanofilament structure of bone ECM. Kim et al. [67] employed sol–gel precursors to realize 1D BG-based electrospun nanofibers, which displayed cytocompatibility and simulated the biomineralization of HAp crystals in simulated body fluids. These BGs-based nanofibers may impart bio-activity as well as afford ECM-like morphological features. Electrospun BGs nanofibers can maintain the bionic nature of the bone ECM as well as leverage bioactive signals for bone tissue repair. The BGs may stimulate/induce the osteogenic differentiation of osteoblasts, inhibit bone resorption and collagen degradation, and promote osteogenic differentiation through relevant signaling pathways. Weng et al. [68] successfully prepared BG nanofibers doped with inorganic metals, such as strontium (Sr) and Cu, by using electrospinning technology (Figure 2A i). The doping of BGs with Sr significantly improved osteogenesis and inhibited osteoclast formation (Figure 2A ii–iv), while doping with the Cu promoted angiogenesis (Figure 2A v–vii). Owing to their ability to mimic the bone microenvironment and release therapeutic ions, these BG nanofibers may hold promise for bone TE. Similarly, Gazquez et al. [69] leveraged electrospinning to fabricate yttria-stabilized zirconia (YSZ) scaffolds by using polyvinylpyrrolidone (PVP) and YSZ precursors. As compared to pure ceramic materials, YSZ ceramic nanofibers exhibited remarkable flexibility as well as promoted the growth of human mesenchymal stromal cells (hMSCs), which may have implications for bone TE owing to the unique combination of the high stiffness and flexibility of resulting scaffolds (Figure 2B).

Recently, bioactive glass nanofiber film consisting of electrostatically spun flexible MgO_2_-doped silica (SiO_2_/MgO) has also been developed [70]. In vitro results revealed good cytocompatibility and bioactivity of these purely inorganic nanofiber membranes, which improved cell proliferation and angiogenesis. The sustained release of silicon and magnesium ions induced antibacterial effects of membranes by modulating the expression of inflammatory factors by stimulating effector cells, thereby promoting healing of infected wounds in a murine allograft model.

#### 2.1.2. Inorganic Oxides-Based Electrospun Nanofibers

A series of inorganic MOs have been spun to afford nanofibrous scaffolds. Silica (SiO_2_) has been widely exploited as a drug delivery carrier as well as a scaffold for TE [71,72,73,74]. In addition, SiO_2_ NPs have been incorporated into polymers to afford electrospun scaffolds [75,76,77]. The release of silicon ions plays an important role in enhancing the biological performance of scaffolds. Electrospun nanofibers solely composed of inorganic SiO_2_ have also been fabricated.

##### Application of Electrospun Nanofibres of Inorganic Oxides Prepared into Hydrogels

Hydrogels are widely used in a variety of biomedical applications due to their inherent ability to retain high water content as well as their good miscibility with a range of natural and synthetic polymers [78,79]. The overall physical and mechanical behavior of hydrogels depends on the underlying internal structure [80]. However, the low mechanical strength and limited functionality of conventional hydrogels adversely affect their use in tissue engineering [81]. Consequently, the combination of hydrogels with electrospun scaffolds may be an effective way to improve inherent limitations associated with hydrogel.

Yang et al. [82] developed nanofibrous hydrogels (NFH) by combining flexible SiO_2_ nanofibers along with ionically-crosslinked alginate. As compared to the hydrogels composed of pristine alginate, NFH exhibited remarkably higher mechanical properties, which were attributable to flexible SiO_2_ nanofibers. The NFH showed a plastic deformation value of only 9.5% after 1000 compression cycles at 50% strain (Figure 3A iii). In addition, the Al-alginate was uniformly wrapped around the surface of the SiO_2_ nanofibers, which further improved the water content of NFH for up to 99.8 wt.% (Figure 3A ii). This highly hydrated and porous nanofibrous structure allowed NFH to maintain a sensitive shape memory recovery function as well as imparted injectability characteristics (Figure 3A i). The combination of highly sensitive responsiveness of NFH with the current and pressure may further open a window of opportunity for research in electrical/pressure-stimulated TE scaffolds (Figure 3A vi. Despite these encouraging results, the biocompatibility of these NFH hydrogels as well as their long-term in vivo implantation yet remains to be explored.

Wang et al. [83] incorporated SiO_2_ nanofiber membranes into CS hydrogels to afford NF/CS composite hydrogels. The addition of SiO_2_ remarkably improved the mechanical properties of hydrogels in contrast to that of cellulose acetate (CA) or polyacrylonitrile (PAN) nanofiber membranes. The obvious effects of SiO_2_ membranes over CA or PAN nanofibers were ascribed to the higher modulus and tetrahedral structure of SiO_2_. The mechanical properties of NF/CS composite hydrogels were also increased with an increase in the SiO_2_ content. Owing to their resemblance to the bone ECM, these composite NF/CS hydrogels not only exhibited higher mineralization and biocompatibility but also promoted the osteogenic differentiation of hMSCs (Figure 4A i,ii), which was even comparable to the hydrogels immobilized with bone morphogenetic protein-2 (BMP2, 50 ng per matrix) (Figure 4A iii). Despite these encouraging in vitro results, in vivo evaluation over a long time period yet remains to be accomplished to better discern the effect of these scaffolds on bone repair.

##### Application of Electrospun Nanofibres of Inorganic Oxides Prepared as Composite Three-Dimensional Scaffolds

Wang et al. [84] developed a 3D fibrous scaffold (SiO_2_NF-CS) constructed from chitosan (CS) layers wrapped around flexible SiO_2_ nanofibers [82]. Unlike the inherent structural fragility of inorganic nanofibers, the SiO_2_NF-CS scaffolds displayed super-resilience in the aqueous environment, showing full recovery to their initial height as well as maintaining an intact porous structure under cyclic compression at strain values for up to 80% (Figure 3B i). These data indicate that the scaffolds can maintain perfect shape recovery properties in the aqueous environment. The inorganic rigid SiO_2_ nanofibers remarkably improved the mechanical properties of the scaffold (Figure 3B iii), which may also have implications for bone TE. The hMSCs maintained >95% cellular activity and a significant growth trend on the SiO_2_NF-CS scaffold (Figure 3B ii). In addition to its good cytocompatibility, SiO_2_NF-CS also induced multi-directional differentiation of hMSCs (Figure 3B iv,v); SiO_2_ promoted the differentiation of MSCs into osteoblasts through enhanced mineral deposition [85]. The implantation of these scaffolds in a cranial defect model in rats further led to significant bone regeneration.

To accurately replicate the composition and stiffness gradients of subchondral bone, Wang et al. fabricated 3D SiO_2_ nanofiber-CS scaffolds (SiO_2_NF-CS), which showed a stiffness gradient attributable to the SiO_2_ nanofiber content. The incorporation of SiO_2_ remarkably improved the mechanical properties of scaffolds. With an increase in the SiO_2_ content into the scaffolds from 0% to 90%, the compressive modulus was increased from 4.5 kPa to 45 kPa, while compressive stress from 4 kPa to 18 kPa, respectively. Intriguingly, the gradient in the SiO_2_ nanofiber content also influenced the stiffness of scaffolds, which promoted the differentiation of hMSCs to chondrocytes and osteoblasts. As can be seen from these data, SiO_2_ plays an important role to not only influence the mechanical properties of scaffolds but also promote the cytocompatibility and differentiation to promote osteochondral regeneration.

Similarly, Wang et al. [86] developed flexible binary SiO_2_-CaO nanofiber membranes to afford 3D SiO_2_-CaO NF/CS scaffolds (Figure 4B i). The 3D SiO_2_-CaO NF/CS scaffolds were further optimized to better mimic the ratio of inorganic/organic components of the bone ECM. The weight ratio of the SiO_2_-CaO to chitosan was 65/35. The addition of flexible SiO_2_-CaO short nanofibers enhanced the stiffness and elasticity of the scaffolds (Figure 4B ii,iii). The hBMSCs co-cultured along with the scaffold showed good biocompatibility and significant biomineralization behavior (Figure 4B iv–vi). These results indicate the potential of the scaffolds to promote osteogenic differentiation in vitro as well as bone repair in a cranial defect model in rats in vivo.

Liu et al. [64] fabricated flexible superelastic organic/inorganic composite aerogel scaffolds consisting of flexible SiO_2_ nanofibers and electrospun poly(L-lactide)/gelatin nanofibers. The scaffolds displayed good elasticity and mechanical strength with a silica content of up to 40% (Figure 4C i). Silicon ions (Si^4+^) were sustainably released from the scaffold for up to 8 weeks, which not only promoted the differentiation of rat bone marrow derived mesenchymal stem cells (rBMSCs) into osteoblasts but also induced angiogenesis by promoting the tube formation of HUVECs in vitro (Figure 4C ii–iii). Implantation of these scaffolds in a cranial defect model in rats simultaneously promoted osteogenesis and angiogenesis (Figure 4C iv). Taken together, these flexible yet sufficiently robust scaffolds may possess good potential for bone TE.

### 2.2. Electrospun Organic–Inorganic Composite Nanofibers

Different types of polymers have been blended along with an array of inorganic NMs to fabricate organic/inorganic hybrids. The most widely explored synthetic biodegradable polymers include polylactic acid (PLA), polycaprolactone (PCL), polyurethane (PU), poly(lactic-co-glycolic acid) (PLGA), and poly(L-lactide-co-ε-caprolactone) (PLCL). On the other hand, natural polymers include collagen, chitosan (CS), bacterial cellulose (BC), silk fibroin (SF), and polysaccharides (PS). It is noteworthy to mention here that synthetic biodegradable polymers can be tailored to afford a range of properties [78,87]. Since either natural or synthetic polymers alone cannot meet the key requirements of ideal scaffolds, they have been blended together to simultaneously realize good biocompatibility and mechanical properties for TE applications. In addition, a wide variety of functional inorganic NMs is blended along with natural and synthetic polymers to further modulate their applications in regenerative medicine and TE. While the advantages of functional inorganic NMs in the context of regenerative medicine and TE have been enumerated in preceding sections, here we have outlined the methodologies adopted to realize functional organic–inorganic nanofibers by employing different types of inorganic NMs. Moreover, the applications of these materials have been discussed in detail for TE applications.

#### 2.2.1. Inorganic–Organic Composite Nanofibers Doped with Bioactive Glass Components

The scaffolds prepared solely by using natural/synthetic polymers may not meet the requirements for the regeneration of targeted tissues/organs or other physico-chemical properties, which may require the introduction of additional components for tissue repair. For instance, scaffolds for musculoskeletal tissue repair should not only simultaneously promote osteogenesis and angiogenesis but should also display appropriate mechanical properties comparable to the targeted tissues, such as ligament, bone, or cartilage. Consequently, functional bioceramics are often added as a reinforcing phase to improve the mechanical properties of the scaffold as well as afford the sustained release of therapeutic ions for functional tissue repair [88]. The interconnected porous structure of the electrospun nanofibers allows for cell adhesion which, when combined with the highly bioactive properties of BG, may further enhance the potential of inorganic–organic nanofibers. The addition of an appropriate content of bioceramics into the polymer solution should not pose an adverse effect on the electrospinning process [89,90].

Labbaf et al. [91] fabricated electrospun PCL/BG fiber membranes. The BG particles improved the cytocompatibility of nanofibers; PCL/BG fibrous membranes showed fast biomineralization after 21 days in simulated body fluid (SBF). Dental pulp stem cells were significantly adhered to PCL/BG fibrous membranes and differentiated to afford a mineralized matrix. Ding et al. [92] designed honeycomb-like polyhydroxybutyrate/poly(ε-caprolactone)/58S sol–gel bioactive glass (PHB/PCL/58S) hybrid scaffolds through self-assembly-driven electrospinning. The addition of 58S BG further improved the hydrophilicity of PHB/PCL membranes, which improved cell adhesion, cell proliferation, and biocompatibility. Similarly, Saraf et al. [93] blended BG NPs along with cellulose acetate to produce organic–inorganic composite nanofibers. Composite scaffolds containing 3% of BG displayed smooth nanofibers. Notably, the addition of BG improved bactericidal properties of scaffolds against gram-negative and gram-positive bacteria. The scaffolds containing BG NPs also promoted wound healing in diabetic rats, which may be ascribed to the antibacterial effects of BG as well as its propensity to induce tissue regeneration.

Gönen et al. [94] synthesized Sr and Cu ions doped BGs and blended them into PCL/Gel to afford nanofibers. The BG NPs induced the deposition of HAp layers as well as promoted the cytocompatibility and enhanced the degradation of fibers (Figure 5A i,ii). The release of Sr and Cu ions induced osteogenic, angiogenic, and antibacterial properties to the scaffolds. Since therapeutic ions play an important role to promote tissue repair, these organic–inorganic nanofibers capable of sustained release of ions may have substantial tissue repair effects (Figure 5A iii). While the incorporation of inorganic ceramic NPs into polymer scaffolds can improve several aspects of the scaffolds, such as biocompatibility, biomineralization, and biodegradability, the inherent stiffness and brittleness of BG NPs still require optimization.

#### 2.2.2. Inorganic–Organic Composite Nanofibres Doped with Hydroxyapatite

Hydroxyapatite (HA, Ca_10_(PO_4_)_6_(OH)_2_) exhibits structure and composition similar to that of natural apatite found in bone tissues and, owing to its high mechanical strength and biocompatibility, it is widely used as a scaffold material for bone TE. However, HA may have several drawbacks as a scaffold material, including a mismatch between the degradation rate and new bone formation as well as the low porosity and plasticity of the scaffolds, which impede its widespread use [95,96].

The HAp can be synthesized by using different types of methods, such as chemical precipitation (CP), hydrothermal (HT), and sol–gel (SG) [97,98]. Wet precipitation (WP) is also commonly used for the synthesis of HAp [99]. However, this approach is limited by diffusivity, whereas the Ginstling–Brounstein (GB) model is suitable for describing diffusion control processes [100]. In addition to traditional methods for HAp synthesis, including CP and SG, alternative approaches, including microwave synthesis, may offer a unique platform to generate HAp with the desired structure and properties [101,102].

In order to overcome the disadvantages associated with the mere use of HAp as well as achieve good osteogenic effect, electrospun scaffolds can be developed by blending HAp along with natural and synthetic polymers. Erdem et al. [103] blended poly(L-propylene-co-ε-caprolactone), Col I, and Ag-doped HAp particles to afford electrospun fibers, which showed good bactericidal properties as well as osteogenic properties. Since native bone exhibits high ceramics content embedded in ECM, recapitulating such features is pivotal for bone TE. Wu et al. [104] blended high content of HAp (~ 60%) along with PCL by using the electrospinning method and systematically studied concentration-dependent effects of HAp on cell infiltration and growth. Similarly, Johari et al. [105] leveraged electrospinning to afford composite nanofibers by blending PCL and fluorinated hydroxyapatite (FHAp). Composites containing 10 wt.% of FHAp NPs afforded smooth and uniform fibers manifesting remarkable biocompatibility and cell adhesion. It is worthy of note that, owing to poor compatibility between polymer matrix and HAp NPs, the resulting scaffolds may exhibit weak mechanical properties and failures, which necessitates further studies. Moreover, owing to the complex nature of the bone defects, while electrospun nanofibers can better fit these defects, precision design of such scaffolds is of considerable significance for their clinical translation. Finally, as the bones are load-bearing tissues, adequate attention should be paid to design scaffolds with optimized mechanical properties to not only provide mechanical support but to also promote reciprocally interaction with the regenerating bone.

#### 2.2.3. Inorganic–Organic Composite Nanofibers Doped with Inorganic Metal Oxides

Similar to the incorporation of bioactive ceramics into scaffold materials, different types of inorganic NPs, such as metals and MOs, have been incorporated into scaffolds. Especially, Mg and magnesium oxide (MgO) have attracted increasing interest in regenerative medicine due to their biocompatibility, biodegradability, and bioactivity. The authors prepared hybrid nanofibers by blending MgO along with a PCL/gel mixture solution [106,107]. Owing to the release of the Mg^2+^, composite membranes containing MgO increased the VEGF production of HUVECs and promoted wound healing in diabetic rat models. The appropriate content of MgO also conferred antimicrobial characteristics to scaffolds to induce wound repair. Similarly, Rijal et al. [108] blended MgO along with PCL and CS. The composite nanofibers exhibited an average fiber diameter in the range of 0.7 to 1.3 μm. The ultimate tensile strength (UTS) of the electrospun membrane was ~3 MPa and Young’s modulus was ~25 MPa (Figure 5C i,ii). The 3T3 fibroblasts cultured on composite nanofibers showed good viability, cell adhesion and cell attachment, thereby indicating the good cytocompatibility of hybrid nanofibers (Figure 5C iii).

In addition, Zn plays a pivotal role for bone structure and metabolism [109]. The Zn also exhibits antibacterial properties and is an essential metal element for cell growth and ECM production. The antibacterial properties of ZnO NPs have been ascribed to the release of zinc ions as well as the production of reactive oxygen species (ROS) [110]. The ZnO NPs have attracted increasing interest in TE due to their excellent antibacterial properties and minimal side effects on human cells. Compared to pure ZnO, hybrid nanofibers containing ZnO micro/nanoparticles may have obvious advantages. To afford antibacterial scaffolds, Zhan et al. [111] functionalized polyacrylonitrile (PAN) nanofibrous membranes with ZnO by a chemical grafting method. The ZnO considerably improved the hydrophilicity of membranes as well as the antibacterial activity and cytocompatibility. Mao et al. [112] developed a multifunctional 3D PLA/gelatin/ZnO nanofiber aerogel scaffolds (Figure 5B i), which showed higher water absorption and permeability than that of the 2D membranes. These nanofibers may have potential to absorb wound exudate while ensuring good gas exchange (Figure 5B ii). The addition of ZnO NPs also showed higher production of ROS, thereby improving the antibacterial properties (Figure 5B iii). The implantation of these hybrid membranes onto infected skin wounds in vivo reduced inflammatory responses and effectively promoted healing (Figure 5B iv). Jaykumar et al. [113] blended ZnO/HAp with PLCL and PLCL/SF. The composite nanofibers showed an average fiber diameter of 139.4 ± 27 nm. Human fetal osteoblasts (hFOB) showed higher cell proliferation (Figure 6A i), osteogenic gene expression, and mineral secretion in scaffolds containing ZnO/HAp than that of the scaffolds devoid of NPs (Figure 6A ii,iii).

#### 2.2.4. Inorganic–Organic Composite Nanofibers Doped with Other Inorganic Components

In addition to the aforementioned inorganic NMs, various types of NMs, including nano silicates (nSi) have also been incorporated into nanofibers. The degradation products of nanosilicates include magnesium ions (Mg^2+^), orthosilicic acids (Si(OH)_4_), and lithium ions. Interaction of inorganic NPs with hMSCs may activate signaling pathways associated with stress responses to induce the osteogenic differentiation of hMSCs [114]. Carrow et al. [115] prepared scaffolds for bone TE by mixing nano-silicates along with poly(ethylene oxide terephthalate)/poly(butylene terephthalate) (PEOT/PBT) copolymers by using 3D printing. The addition of 2D nanosilicate inhibited the degradation of the copolymer, thereby ensuring the stability of the scaffold under physiological conditions (Figure 6B i). Noticeably, nanosilicates can induce the osteogenic differentiation of hMSCs even in the absence of osteo-inductive agents. Consequently, hMSCs are more likely to proliferate and upregulate bone-associated proteins and promote biomineralization on scaffolds containing nanosilicates (Figure 6B ii–iv). Consequently, the addition of nanosilicates provides an alternative approach to develop multifunctional scaffolds for BTE.

Wang et al. [116] successfully prepared organic/inorganic nanofiber membranes containing different content of nSi (i.e., PCL—0%nSi, PCL—1%nSi, PCL—5%nSi and PCL—10%nSi) and assessed their potential for in vivo ectopic osteogenesis in rats. At low nSi content, the mechanical properties of the nanofibers were improved (Figure 6C i), however, nSi adversely affected the cytocompatibility of scaffolds. The nanofibers containing nSi may not only promote biomineralization in vitro but may also induce ectopic bone formation in vivo as compared to pure PCL scaffolds (Figure 6C ii,iii).

## 3. Conclusions and Future Outlook

As the field of TE continues to evolve, nanofiber scaffolds fabricated either by using natural or synthetic polymers may be insufficient to induce a tissue-specific reparative effect. Commonly used synthetic polymers show excellent mechanical properties and biodegradability; however, poor biocompatibility may limit the application prospects of resulting scaffolds in TE. Likewise, while natural polymers exhibit good biocompatibility and biodegradability, as well as mimic ECM for microstructure and tethered ligands, they lack mechanical properties as required for musculoskeletal TE and related disciplines. It is noteworthy to mention here that while designing scaffold materials for TE, the target or gold-standard mechanical properties should be taken into consideration, which may include, for instance, strength, modulus, fatigue/viscoelastic creep properties as well as thermal expansion, chemical stability/resistance against ageing etc. Consequently, hybrid scaffolds consisting of natural and synthetic polymers may have great application prospects to appropriately tailor the properties of hybrid scaffolds. For bone TE, HA composites have been shown to exhibit compressive strength between 2 to 230 MPa and modulus of elasticity in the range of 0.05 to 30 GPa, thereby matching the mechanical properties of bone tissues. In addition, the scaffolds displaying pore sizes above 50 μm and degradation period between 2 to 6 months have been shown to facilitate cellular infiltration and angiogenesis for bone TE [95]. As mentioned above, despite good potential of hybrid scaffolds solely fabricated by using natural/synthetic polymers, additional candidates, such as inorganic NMs may be harnessed to further ameliorate the limitations associated with these scaffolds as well as confer additional bioactivity to scaffolds. Different types of inorganic NMs, such as BGs, metal NPs, and MOs may be blended along with natural/synthetic polymers as alternative materials for a wide range of scaffolds by using different techniques (Table 1). Organic–inorganic composite scaffolds may help realize the dual functional synergies of organic and inorganic materials to achieve better repair results. While we have mainly discussed composite nanofiber scaffolds consisting of natural/synthetic polymers and inorganic NMs prepared by using electrospinning methods, other types of fabrication techniques may also be harnessed to fully exploit the therapeutic effect of inorganic NMs.

Additionally, owing to the uniqueness of different types of tissues in terms of their physico-chemical properties, it is of considerable significance to better predict or model the properties of scaffolds as well as their performance following transplantation in vivo. It is also of interest to note that some recent studies have shown that the modeling can provide a convenient way to predict the properties of polymers and polymer composites. For example, modelling of the tensile strength of polymers to predict the mechanical properties of binary polymer blends of different compositions [117]. A temperature- and damage-dependent tensile strength (TDDTS) model for polymer fiber/composites may enable analysis of the factors influencing the tensile properties and damage evolution with temperature [118]. Similarly, modeling studies may help predict minimum strength and dimensional effects of 3D printed polymers by using fused deposition modeling and injection moulding [119]. Consequently, these modeling and machine learning tools may be instrumental to simulate and predict the properties of inorganic NMs or organic/inorganic composites, including strength, modulus, fatigue/viscoelastic creep properties as well as thermal expansion, chemical stability/resistance against ageing, etc. Consequently, such screening tools of scaffolds and biomaterials may help design advanced TE platforms to accurately recapitulate the features of ECM as well as predict the performance of transplants in vivo, thereby holding great promise for future biomaterials discovery.

Since inorganic NMs exhibit good physico-chemical properties and bioactivity, they can significantly improve the performance of scaffolds and may also have a direct impact on the growth of different types of cells. Since inorganic ions have been shown to be the important regulators of angiogenesis and osteogenesis, precisely-designed inorganic NMs may be harnessed to simultaneously promote angiogenesis and antibacterial effects as well as scavenge ROS [120] for tissue repair. The strategy of introducing inorganic NMs may further widen the application prospect of biomaterials. While an array of inorganic NMs have already been harnessed to afford functional scaffolds for TE, further research is warranted to better delineate their effect to promote tissue repair. Similarly, whereas composite nanofibers doped with inorganic NMs have been widely harnessed for musculoskeletal tissue repair, their application for other injury/tissue types, such as skin, heart, ischemia, muscle, and nerve may further be exploited. The precise design of inorganic NMs focusing the particular requirement of the therapeutic ions needed for the targeted tissue types is further warranted. Equally important, while different types of inorganic NMs can first be synthesized and then be incorporated into nanofiber scaffolds, their distribution, appropriate content, and biocompatibility should be carefully considered. The inclusion of inappropriate amounts of inorganic components may lead to rapid release inducing cytotoxicity, which requires a careful attention. In addition, inorganic NPs need to be compounded with organic polymer materials for the preparation of electrospun nanofibers, and there are many limitations on the amount and size of doping, which need to be carefully considered for future applications during the fabrication of scaffolds. On the contrary, in-situ synthesis of inorganic NMs during electrospinning or their incorporation through reactive electrospinning based techniques may not only shorten the fabrication steps but may also afford advanced regenerative medicine and TE platforms. These approaches may further help develop off-the-shelf available nanofiber platforms for a range of injuries and defects.

In addition, a series of flexible nanofibrous membranes of purely inorganic NMs (e.g., SiO_2_ nanofiber membranes) have been prepared by electrospinning techniques in recent studies and have shown good results in specific tissue repair processes. However, there are still several issues, which need to be carefully addressed for the translation of inorganic NMs-based fibers for TE applications. There are perpetuating challenges about the technological progress of the synthesis of inorganic NMs as well as their hybrids with natural and synthetic polymers by electrospinning method, which requires careful attention for scalability, uniformity, and homogeneity for commercialization and clinical applications [121]. Electrospun membranes of inorganic MOs may show weak mechanical properties and fragility, presumably due to the high content of inorganic NMs, which may impede their application in TE. In future studies, the preparation of other kinds of inorganic MOs nanofiber membranes should be further explored, and pure inorganic nanofiber membranes with good flexibility, such as the SiO_2_ nanofibers mentioned above, should be prepared by adjusting the preparation process, such as calcination temperature, electrostatic spinning parameters and material ratio, etc. Similarly, release of therapeutic ions from inorganic NMs has been shown to promote tissue repair. Specifically, mesoporous SiO_2_, Mg(OH)_2_, cobalt (Co), copper (Cu) and BGs based scaffolds have been put forwarded and shown to stimulate tissue repair process as well as afford microbial protection and anti-inflammatory properties through release of different types of ions [70,120,122,123,124]. Consequently, release kinetics of therapeutic ions should be carefully considered to possibly predict the safe window for their therapeutic effects while safeguarding cells and tissues in vivo from toxicity risks; the overproduction of therapeutic ions may adversely affect cell viability and tissue repair process, which warrants further detailed studies.

To conclude, much progress has been made in the field of TE in the production of scaffolds by using different types of inorganic NMs, and many studies have highlighted the potential of these composite nanofibers in controlling specific tissue functions. Nevertheless, long-term therapy with inorganic-bound scaffolds in TE applications still requires long-term pre-clinical studies.

## Data Availability

Not applicable.

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
