# Peer review of "Electrospinning Inorganic Nanomaterials to Fabricate Bionanocomposites for Soft and Hard Tissue Repair"

_nanomaterials, 2023, doi:10.3390/nano13010204_

Round 1

Reviewer 1 Report

Ms. ID: nanomaterials-2098843

Title: Recent Advancements on Inorganic Electrospun Nanofiber Scaffolds for Tissue Engineering Applications

Reviewer comments

The generic part of this review is relatively short, which is a good thing for the brevity of the presentation. Considering the long-standing position of e-spinning as a scaffold preparation method for tissue engineering, many papers have been published that discuss various aspects. Focused reviews are needed to cover specific interest areas, and this review is a good example of that approach. Inorganic e-spun scaffolds are an exotic class that departs from their polymer-based counterparts in many aspects, despite the fact that examples of standalone inorganic scaffolds are scarce in the available literature. Most of the work in this area, or rather the route of synthesis, leads to the preparation of hybrid polymer-inorganic NPs materials. The NPs are distributed in the volume of fibers, which usually leads to some unusual properties of these scaffolds, such as improved osteoconductivity. In general, these scaffolds are predominantly used in the tissue engineering of various bone structures, which is rather intuitive.

This relatively short review is also well-illustrated, with a carefully selected set of figures borrowed from published papers (with permission from the publisher). Overall, the review is concise, focused and well-written. This reviewer doesn’t have any other concerns.

Specific comments

Line 119: “In addition, an array of materials has been employed for the mass production of continuous nanofibers with high specific surface area and suitable mechanical properties [37, 38].” High specific surface area (SSA) is not a correct phrase to characterize nanofibers. It is a common misconception, with the origin unknown to this reviewer, that propagates in scientific literature. There are numerous materials that have much higher SSA thanks to internal porosity, which e-spun fibers usually don’t have (hundreds-to-thousands sq. meters vs a few sq. meters).

Reviewer 2 Report

The review is dedicated to the electrospining of inorganic nanofibered materials for tissue engineering. The topic is interesting and there's not a lot of review papers concentrating on this subject. The manuscript is comprehensive enough and logically organized, so I think it can be published in the present form.

The only comment I have - there are many abbreviations in the text. Some of them are not common, so I suggest authors to make a list of used abbreviations and put it somewhere in the beginning of the manuscript, otherwise reader spends a lot of time trying to find the first mention of abbreviation.

Reviewer 3 Report

The manuscript "Recent Advancements on Inorganic Electrospun Nanofiber Scaffolds for Tissue Engineering Applications" is a good-quality review article in the field of Tissue Engineering. I have some comments, which should be addressed by the team:

[1] There are inconsistencies present in the formatting of the documents, e.g., see pages 1-2 from line 38 to line 49. There are similar inconsistencies also later in the document. Please address this issue.

[2] Hydroxylapatite (HAp) is discussed, however, since this is a review article on recent advancements, there are some missing references about advancements in HAp and its synthesis -- for instance, 10.5277/ABB-01196-2018-02 and https://doi.org/10.1515/chem-2018-0011 .

[3] Discuss recent HAp advancements in terms of electrospinning applications -- how do these relate.

[4] The authors state "While commonly used synthetic polymers show excellent mechanical properties and biodegradability, the poor biocompatibility of these polymers may limit their applications. On the other hand, while natural polymers show good biocompatibility and bio-degradability, they lack mechanical properties as may be required for musculoskeletal TE and related disciplines.". In respect to this statement, what is the balance, the "golden middle" in terms of polymer properties that is required -- it would be very valuable, if the authors could provide the range of properties that is needed for a succesful polymer implementation in TE applications, such as Strength, Modulus, Fatigue/viscoelastic creep properties as well as thermal expansion, chemical stability / resistance against ageing etc. In the recent advancements, the modelling tools are enabling the predictions of properties and ageing of polymer and polymer composite properties, and this has been recently recounted in the litretature to a quite detailed extent. Please take a look into the recent advancements on the modelling/prediction of such properties, and discuss how this could be used in the advancement in TE.

I recommend this article for publication after the issues mentioned above have been addressed.

// Minor Revision

Reviewer 4 Report

I reviewed the article entitled „Recent Advancements on Inorganic Electrospun Nanofiber Scaffolds for Tissue Engineering Applications“. The topic has already been reviewed in following publications:

Wang Z., Shi X. Construction of drug-loaded electrospun organic/inorganic hybrid nanofibers for biomedical applications. Materials China, 2014. DOI: 10.7502/j.issn.1674-3962.2014.11.03

Huang W., Xiao Y., Shi X. Construction of Electrospun Organic/Inorganic Hybrid Nanofibers for Drug Delivery and Tissue Engineering Applications. Advanced Fiber Materials, 2019. DOI: 10.1007/s42765-019-00007-w

I recommend the authors to explain their novelty of contribution compared to published results.

The topic of the review is mostly focused on composite organic / inorganic hybrid materials, therefore the title of the article should be rewritten. There were investigated also pure inorganic nanofibers for tissue engineering applications that are not mentioned within the text.

In the beginning of the review, summarizing Table with type of incorporated inorganic materials should be added.

Inorganic materials should be incorporated by various methods which shoulh also be described within the text.

Figure 1 describes utilization of needle electrospinning. What about needleless electrospinning which is much more productive?

Some of the materials are fabricated by combination of methods (hydrogels, 3D printing). These composite materials should be described separately.

The text contains many references bringing deep inside into hybrid organic/inorganic materials. However, I recommend the authors to rething the construction of the text making it more attractive and „clear“ to readers based on above mentioned comments.

Last but not least, disadvantages of such aterials should be more deeply discussed such as technological issues regarding fabrication of such materials, release of incorporated materials and its possible cytotoxicity etc.

Author Response

请参阅附件。

Round 2

Reviewer 4 Report

The authors significantly improved the manucript and I recommend it for publication.